# Differentiable Reasoning over a Virtual Knowledge Base

**Bhuwan Dhingra**[1]*, **Manzil Zaheer**[2], **Vidhisha Balachandran**[1],
**Graham Neubig**[1], **Ruslan Salakhutdinov**[1], **William W. Cohen**[2]
[1] School of Computer Science, Carnegie Mellon University
[2] Google Research
{bdhingra, vbalacha, gneubig, rsalakhu}@cs.cmu.edu
{manzilzaheer, wcohen}@google.com

## Abstract

We consider the task of answering complex multi-hop questions using a corpus as a *virtual knowledge base (KB)*. In particular, we describe a neural module, DrKIT, that traverses textual data like a KB, softly following paths of relations between *mentions* of entities in the corpus. At each step the module uses a combination of sparse-matrix TFIDF indices and a maximum inner product search (MIPS) on a special index of contextual representations of the mentions. This module is differentiable, so the full system can be trained end-to-end using gradient based methods, starting from natural language inputs. We also describe a pretraining scheme for the contextual representation encoder by generating hard negative examples using existing knowledge bases. We show that DrKIT improves accuracy by 9 points on 3-hop questions in the MetaQA dataset, cutting the gap between text-based and KB-based state-of-the-art by 70%. On HotpotQA, DrKIT leads to a 10% improvement over a BERT-based re-ranking approach to retrieving the relevant passages required to answer a question. DrKIT is also very efficient, processing 10-100x more queries per second than existing multi-hop systems.[1]

## 1 Introduction

Large knowledge bases (KBs), such as Freebase and WikiData, organize information around entities, which makes it easy to reason over their contents. For example, given a query like *"When was the Grateful Dead's lead singer born?"*, one can identify the entity *Grateful Dead* and the path of relations *LeadSinger*, *BirthDate* to efficiently extract the answer—provided that this information is present in the KB. Unfortunately, KBs are often incomplete (Min et al., 2013). While relation extraction methods can be used to populate KBs, this process is inherently error-prone, expensive and slow.

Advances in open-domain QA (Moldovan et al., 2002; Yang et al., 2019) suggest an alternative—instead of performing relation extraction, one could treat a large corpus as a virtual KB by answering queries with spans from the corpus. This ensures facts are not lost in the relation extraction process, but also poses challenges. One challenge is that it is relatively expensive to answer questions using QA models which encode each document in a query-dependent fashion (Chen et al., 2017; Devlin et al., 2019)—even with modern hardware (Strubell et al., 2019; Schwartz et al., 2019). The cost of QA is especially problematic for certain complex questions, such as the example question above. If the passages stating that *"Jerry Garcia was the lead singer of the Grateful Dead"* and *"Jerry Garcia was born in 1942"* are far apart in the corpus, it is difficult for systems that retrieve and read a single passage to find an answer—even though in this example, it might be easy to answer the question after the relations were explicitly extracted into a KB. More generally, complex questions involving sets of entities or paths of relations may require aggregating information from multiple documents, which is expensive.

One step towards efficient QA is the recent work of Seo et al. (2018; 2019) on phrase-indexed question answering (PIQA), in which spans in the text corpus are associated with question-independent

---

*Part of this work was done during an internship at Google.
[1] Code available at http://www.cs.cmu.edu/~bdhingra/pages/drkit.html

contextual representations and then indexed for fast retrieval. Natural language questions are then answered by converting them into vectors that are used to perform maximum inner product search (MIPS) against the index. This can be done efficiently using approximate algorithms (Shrivastava & Li, 2014). However, this approach cannot be directly used to answer complex queries, since by construction, the information stored in the index is about the local context around a span—it can only be used for questions where the answer can be derived by reading a single passage.

This paper addresses this limitation of phrase-indexed question answering. We introduce an efficient, end-to-end differentiable framework for doing complex QA over a large text corpus that has been encoded in a query-independent manner. Specifically, we consider "multi-hop" complex queries which can be answered by repeatedly executing a "soft" version of the operation below, defined over a set of entities $X$ and a relation $R$:

$$Y = X.\text{follow}(R) = \{x' : \exists x \in X \text{ s.t. } R(x, x') \text{ holds}\}$$

In past work soft, differentiable versions of this operation were used to answer multi-hop questions against an explicit KB (Cohen et al., 2019). Here we propose a more powerful neural module which approximates this operation against an indexed corpus (a virtual KB). In our module, the input $X$ is a sparse-vector representing a weighted set of entities, and the relation $R$ is a dense feature vector, e.g. a vector derived from a neural network over a natural language query. $X$ and $R$ are used to construct a MIPS query used for retrieving the top-$K$ spans from the index. The output $Y$ is another sparse-vector representing the weighted set of entities, aggregated over entity mentions in the top-$K$ spans. We discuss pretraining schemes for the index in §2.3.

For multi-hop queries, the output entities $Y$ can be recursively passed as input to the next iteration of the same module. The weights of the entities in $Y$ are differentiable w.r.t the MIPS queries, which allows end-to-end learning *without* any intermediate supervision. We discuss an implementation based on sparse-matrix-vector products, whose runtime and memory depend *only* on the number of spans $K$ retrieved from the index. This is crucial for scaling up to large corpora, providing up to 15x faster inference than existing state-of-the-art multi-hop and open-domain QA systems. The system we introduce is called DrKIT (for Differentiable Reasoning over a Knowledge base of Indexed Text). We test DrKIT on the MetaQA benchmark for complex question answering, and show that it improves on prior text-based systems by 5 points on 2-hop and 9 points on 3-hop questions, reducing the gap between text-based and KB-based systems by 30% and 70%, respectively. We also test DrKIT on a new dataset of multi-hop slot-filling over Wikipedia articles, and show that it outperforms DrQA (Chen et al., 2017) and PIQA (Seo et al., 2019) adapted to this task. Finally, we apply DrKIT to multi-hop information retrieval on the HotpotQA dataset (Yang et al., 2018), and show that it significantly improves over a BERT-based reranking approach, while being 10x faster.

## 2 Differentiable Reasoning over a KB of Indexed Text

We want to answer a question $q$ using a text corpus as if it were a KB. We start with the set of entities $z$ in the question $q$, and would ideally want to follow relevant outgoing relation edges in the KB to arrive at the answer. To simulate this behaviour on text, we first expand $z$ to set of co-occurring mentions $m$ (say using TFIDF). Not all of these co-occurring mentions are relevant for the question $q$, so we train a neural network which filters the mentions based on a relevance score of $q$ to $m$. Then we can aggregate the resulting set of mentions $m$ to the entities they refer to, ending up with an ordered set $z'$ of entities which are answer candidates, very similar to traversing the KB. Furthermore, if the question requires more than one hop to answer, we can repeat the above procedure starting with $z'$. This is depicted pictorially in Figure 1.

We begin by first formalizing this idea in a probabilistic framework in §2.1. In §2.2, we describe how the expansion of entities to mentions and the filtering of mentions can be performed efficiently, using sparse-matrix products and MIPS algorithms (Johnson et al., 2017). Lastly we discuss a pretraining scheme for constructing the mention representations in §2.3.

**Notation:** We denote the given corpus as $\mathcal{D} = \{d_1, d_2, \ldots\}$, where each $d_k = (d_k^1, \ldots, d_k^{L_k})$ is a sequence of tokens. We start by running an entity linker over the corpus to identify mentions of a fixed set of entities $\mathcal{E}$. Each mention $m$ is a tuple $(e_m, k_m, i_m, j_m)$ denoting that the text span $d_{k_m}^{i_m}, \ldots, d_{k_m}^{j_m}$ in document $k_m$ mentions the entity $e_m \in \mathcal{E}$, and the collection of all mentions in the corpus is denoted as $\mathcal{M}$. Note that typically $|\mathcal{M}| \gg |\mathcal{E}|$.

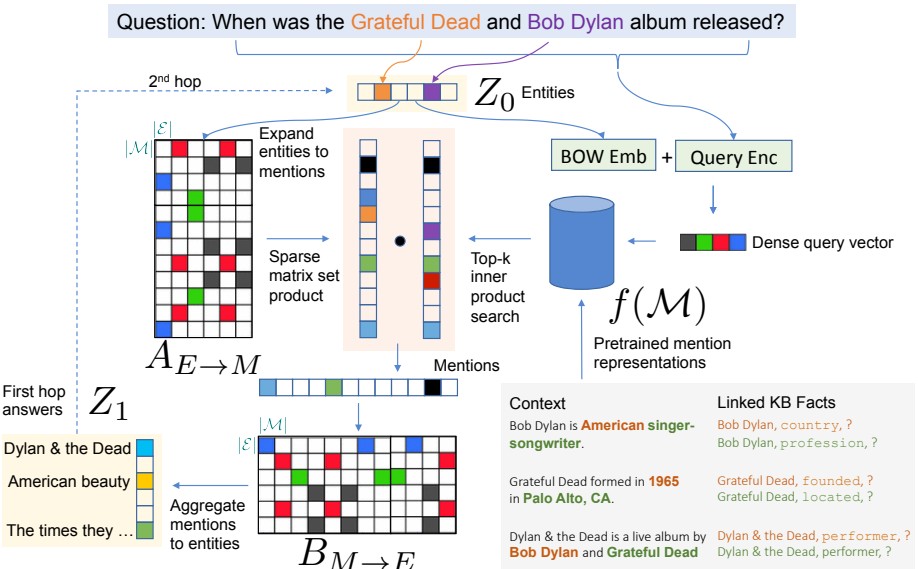

Figure 1: DrKIT answers multi-hop questions by iteratively mapping an input set of entities $X$ (*The Grateful Dead*, *Bob Dylan*) to an output set of entities $Y$ (*Dylan & the Dead*, *American beauty*, ...) which are related to any input entity by some relation $R$ (*album by*).

## 2.1 DIFFERENTIABLE MULTI-HOP REASONING

We assume a *weakly supervised* setting where during training we only know the final answer entities $a \in \mathcal{E}$ for a $T$-hop question. We denote the latent sequence of entities which answer each of the intermediate hops as $z_0, z_1, \ldots, z_T \in \mathcal{E}$, where $z_0$ is mentioned in the question, and $z_T = a$. We can recursively write the probability of an intermediate answer as:

$$\Pr(z_t|q) = \sum_{z_{t-1} \in \mathcal{E}} \Pr(z_t|q, z_{t-1}) \Pr(z_{t-1}|q) \tag{1}$$

Here $\Pr(z_0|q)$ is the output of an entity linking system over the question, and $\Pr(z_t|q, z_{t-1})$ corresponds to a single-hop model which answers the $t$-th hop, *given* the entity from the previous hop $z_{t-1}$, by following the appropriate relation. Eq. 1 models reasoning over a chain of latent entities, but when answering questions over a text corpus, we must reason over entity *mentions*, rather than entities themselves. Hence $\Pr(z_t|q, z_{t-1})$ needs to be aggregated over all mentions of $z_t$, which yields

$$\Pr(z_t|q) = \sum_{m \in \mathcal{M}} \sum_{z_{t-1} \in \mathcal{E}} \Pr(z_t|m) \Pr(m|q, z_{t-1}) \Pr(z_{t-1}|q) \tag{2}$$

The interesting term to model in the above equation is $Pr(m|q, z_{t-1})$, which represents the relevance of mention $m$ given the question and entity $z_{t-1}$. Following the analogy of a KB, we first expand the entity $z_{t-1}$ to co-occuring mentions $m$ and use a learned scoring function to find the relevance of these mentions. Formally, let $F(m)$ denote a TFIDF vector for the document containing $m$, $G(z_{t-1})$ be the TFIDF vector of the *surface form* of the entity from the previous hop, and $s_t(m, z, q)$ be a learnt scoring function (different for each hop). Thus, we model $\Pr(m|q, z_{t-1})$ as

$$\Pr(m|q, z_{t-1}) \propto \underbrace{\mathbb{1}\{G(z_{t-1}) \cdot F(m) > \epsilon\}}_{\text{expansion to co-occurring mentions}} \times \underbrace{s_t(m, z_{t-1}, q)}_{\text{relevance filtering}} \tag{3}$$

Another equivalent way to look at our model in Eq. 3 is that the second term retrieves mentions of the correct *type* requested by the question in the $t$-th hop, and the first term filters these based on co-occurrence with $z_{t-1}$. When dealing with a large set of mentions $m$, we will typically retain only the top-$K$ relevant mentions. We will show that this joint modelling of co-occurrence and relevance is important for good performance, as was also observed by Seo et al. (2019).

The other term left in Eq. 2 is $\Pr(z|m)$, which is 1 if mention $m$ refers to the entity $z$ else 0, based on the entity linking system. In general, to compute Eq. 2 the mention scoring of Eq. 3 needs to

be evaluated for all latent entity and mention pairs, which is prohibitively expensive. However, by restricting $s_t$ to be an inner product we can implement this efficiently (§2.2).

To highlight the differentiability of the proposed overall scheme, we can represent the computation in Eq. 2 as matrix operations. We pre-compute the TFIDF term for all entities and mentions into a *sparse* matrix, which we denote as $A_{E \to M}[e, m] = \mathbb{1}\left(G(e) \cdot F(m) > \epsilon\right)$. Then entity expansion to co-occuring mentions can be done using a sparse-matrix by sparse-vector multiplication between $A_{E \to M}$ and $z_{t-1}$. For the relevance scores, let $\mathbb{T}_K(s_t(m, z_{t-1}, q))$ denote the top-$K$ relevant mentions encoded as a *sparse* vector in $\mathbb{R}^{|\mathcal{M}|}$. Finally, the aggregation of mentions to entities can be formulated as multiplication with another sparse-matrix $B_{M \to E}$, which encodes *coreference*, i.e. mentions corresponding to the same entity. Putting all these together, using $\odot$ to denote element-wise product, and defining $Z_t = [\Pr(z_t = e_1 | q); \ldots; \Pr(z_t = e_{|\mathcal{E}|} | q)]$, we can observe that for large $K$ (i.e., as $K \to |\mathcal{M}|$), Eq. 2 becomes equivalent to:

$$Z_t = \mathrm{softmax}\left(\left[Z_{t-1}^T A_{E \to M} \odot \mathbb{T}_K(s_t(m, z_{t-1}, q))\right] B_{M \to E}\right). \tag{4}$$

Note that *every operation in above equation is differentiable and between sparse matrices and vectors*: we will discuss efficient implementations in §2.2. Further, the number of non-zero entries in $Z_t$ is bounded by $K$, since we filtered (the element-wise product in Eq. 4) to top-$K$ relevant mentions among TFIDF based expansion and since each mention can only point to a single entity in $B_{M \to E}$. This is important, as it prevents the number of entries in $Z_t$ from exploding across hops (which might happen if, for instance, we added the relevance and TFIDF scores instead).

We can view $Z_{t-1}, Z_t$ as weighted multisets of entities, and $s_t(m, z, q)$ as implicitly selecting mentions which correspond to a relation $R$. Then Eq. 4 becomes a differentiable implementation of $Z_t = Z_{t-1}.\mathrm{follow}(R)$, i.e. mimicking the graph traversal in a traditional KB. We thus call Eq. 4 a *textual follow operation*.

**Training and Inference.** The model is trained end-to-end by optimizing the cross-entropy loss between $Z_T$, the weighted set of entities after $T$ hops, and the ground truth answer set $A$. We use a temperature coefficient $\lambda$ when computing the softmax in Eq, 4 since the inner product scores of the top-$K$ retrieved mentions are typically high values, which would otherwise result in very peaked distributions of $Z_t$. We also found that taking a *maximum* over the mention set of an entity $M_{z_t}$ in Eq. 2 works better than taking a sum. This corresponds to optimizing only over the most confident mention of each entity, which works for corpora like Wikipedia that do not have much redundancy. A similar observation was made by Min et al. (2019) in weakly supervised settings.

## 2.2 EFFICIENT IMPLEMENTATION

**Sparse TFIDF Mention Encoding.** To compute the sparse-matrix $A_{E \to M}$ for entity-mention expansion in Eq. 4, the TFIDF vectors $F(m)$ and $G(z_{t-1})$ are constructed over unigrams and bigrams, hashed to a vocabulary of $16M$ buckets. While $F$ computes the vector from the whole passage around $m$, $G$ only uses the surface form of $z_{t-1}$. This corresponds to retrieving all mentions in a document using $z_{t-1}$ as the query. We limit the number of retrieved mentions per entity to a maximum of $\mu$, which leads to a $|\mathcal{E}| \times |\mathcal{M}|$ sparse-matrix.

**Efficient Entity-Mention expansion.** The expansion from a set of entities to mentions occurring around them can be computed using the sparse-matrix by sparse-vector product $Z_{t-1}^T A_{E \to M}$. A simple lower bound for multiplying a sparse $|\mathcal{E}| \times |\mathcal{M}|$ matrix, with maximum $\mu$ non-zeros in each row, by a sparse $|\mathcal{E}| \times 1$ vector with $K$ non-zeros is $\Omega(K\mu)$. Note that this lower bound is independent of the size of matrix $A_{E \to M}$, or in other words independent of the number of entities or mentions. To attain the lower bound, the multiplication algorithm must be vector driven, because any matrix-driven algorithms need to at least iterate over all the rows. Instead we *slice* out the relevant rows from $A_{E \to M}$. To enable this our solution is to represent the sparse-matrix $A_{E \to M}$ as two row-wise *lists of variable-sized lists* of the indices and

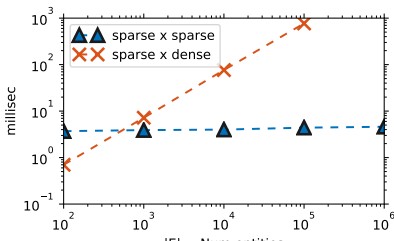

Figure 2: Runtime on a single K80 GPU when using ragged representations for implementing sparse-matrix vector product, vs the default sparse-matrix times dense vector product available in TensorFlow. $|\mathcal{E}| > 10^5$ leads to OOM for the latter.

values of the non-zero elements, respectively. This results in a "ragged" representation of the matrix (tf.RaggedTensors, 2018) which can be easily sliced corresponding to the non-zero entries in the vector in $O(\log |\mathcal{E}|)$ time. We are now left with $K$ sparse-vectors with at most $\mu$ non-zero elements in each. We can add these $K$ sparse-vectors weighted by corresponding values from the vector $Z_{t-1}^T$ in $O(K \max\{K, \mu\})$ time. Moreover, such an implementation is feasible with deep learning frameworks such as TensorFlow. We tested the scalability of our approach by varying the number of entities for a fixed density of mentions $\mu$ (from Wikipedia). Figure 2 compares our approach to the default sparse-matrix times dense-vector product available in TensorFlow.

**Efficient top-$K$ mention relevance filtering:** To make computation of Eq. 4 feasible, we need an efficient way to get top-$K$ relevant mentions related to an entity in $z_{t-1}$ for a given question $q$, without enumerating all possibilities. A key insight is that by restricting the scoring function $s_t(m, z_{t-1}, q)$ to an inner product, we can easily approximate a parallel version of this computation, across all mentions $m$. To do this, let $f(m)$ be a dense encoding of $m$, and $g_t(q, z_{t-1})$ be a dense encoding of the question $q$ for the $t$-th hop, both in $\mathbb{R}^p$ (the details of the dense encoding is provided in next paragraph), then the scoring function $s_t(m, z_{t-1}, q)$ becomes

$$s_t(m, z_{t-1}, q) \propto \exp\{f(m) \cdot g_t(q, z_{t-1})\}, \tag{5}$$

which can be computed in parallel by multiplying a matrix $f(\mathcal{M}) = [f(m_1); f(m_2); \ldots]$ with $g_t(q, z_{t-1})$. Although this matrix will be very large for a realistic corpus, since eventually we are only interested in the top-$K$ values, we can use an approximate algorithm for Maximum Inner Product Search (MIPS) (Andoni et al., 2015; Shrivastava & Li, 2014) to find the $K$ top-scoring elements. The complexity of this filtering step using MIPS is roughly $O(Kp \operatorname{polylog}|\mathcal{M}|)$.

**Mention and Question Encoders.** Mentions are encoded by passing the passages they are contained in through a BERT-large (Devlin et al., 2019) model (trained as described in §2.3). Suppose mention $m$ appears in passage $d$, starting at position $i$ and ending at position $j$. Then $f(m) = W^T[H_i^d; H_j^d]$, where $H^d$ is the sequence of embeddings output from BERT, and $W$ is a linear projection to size $p$. The queries are encoded with a smaller BERT-like model: specifically, they are tokenized with WordPieces (Schuster & Nakajima, 2012), appended to a special `[CLS]` token, and then passed through a $4$-layer Transformer network (Vaswani et al., 2017) with the same architecture as BERT, producing an output sequence $H^q$. The $g_t$ functions are defined similarly to the BERT model used for SQuAD-style QA. For each hop $t = 1, \ldots, T$, we add two additional Transformer layers on top of $H^q$, which will be trained to produce MIPS queries from the `[CLS]` encoding; the first added layer produces a MIPS query $H_{st}^q$ to retrieve a start token, and the second added layer a MIPS query $H_{en}^q$ to retrieve an end token. We concatenate the two and define $\tilde{g}_t(q) = V^T[H_{st}^q; H_{en}^q]$. Finally, to condition on current progress we add the embeddings of $z_{t-1}$. Specifically, we use entity embeddings $E \in \mathbb{R}^{|\mathcal{E}| \times p}$, to construct an average embedding of the set $Z_{t-1}$, as $Z_{t-1}^T E$, and define $g_t(q, z_{t-1}) \equiv \tilde{g}_t(q) + Z_{t-1}^T E$. To avoid a large number of parameters in the model, we compute the entity embeddings as an average over the word embeddings of the tokens in the entity's surface form. The computational cost of the question encoder $g_t(q)$ is $O(p^2)$.

Thus our total computational complexity to answer a query is $\tilde{O}(K \max\{K, \mu\} + Kp + p^2)$ (almost independent to number of entities or mentions!), with $O(\mu|\mathcal{E}| + p|\mathcal{M}|)$ memory to store the precomputed matrices and mention index.[2]

## 2.3 PRETRAINING THE INDEX

Ideally, we would like to train the mention encoder $f(m)$ end-to-end using labeled QA data only. However, this poses a challenge when combined with approximate nearest neighbor search—since after every update to the parameters of $f$, one would need to recompute the embeddings of all mentions in $\mathcal{M}$. We thus adopt a staged training approach: we first pre-train a mention encoder $f(m)$, then compute and index embeddings for all mentions once, keeping these embeddings fixed when training the downstream QA task. Empirically, we observed that using BERT representations "out of the box" do not capture the kind of information our task requires (Appendix C), and thus, pretraining the encoder to capture better mention understanding is a crucial step.

One option adopted by previous researchers (Seo et al., 2018) is to fine-tune BERT on SQuAD (Rajpurkar et al., 2016). However, SQuAD is limited to only 536 articles from Wikipedia, leading to

---

[2]Following standard convention, in $\tilde{O}$ notation we suppress poly $\log$ dependence terms.

| MetaQA | | | | WikiData | | | |
|---|---|---|---|---|---|---|---|
| **Model** | **1hop** | **2hop** | **3hop** | **Model** | **1hop** | **2hop** | **3hop** |
| DrQA (ots) | 0.553 | 0.325 | 0.197 | DrQA (ots, cascade) | 0.287 | 0.141 | 0.070 |
| | | | | PIQA (ots, cascade) | 0.240 | 0.118 | 0.064 |
| KVMem† | 0.762 | 0.070 | 0.195 | | | | |
| GraftNet† | 0.825 | 0.362 | 0.402 | PIQA (pre, cascade) | 0.670 | 0.369 | 0.182 |
| PullNet† | 0.844 | 0.810 | 0.782 | DrKIT (pre, cascade) | 0.816 | 0.404 | 0.198 |
| DrKIT (e2e) | 0.844 | 0.860 | **0.876** | DrKIT (e2e) | **0.834** | **0.469** | **0.244** |
| DrKIT (strong sup.) | **0.845** | **0.871** | 0.871 | –BERT index | 0.643 | 0.294 | 0.165 |

Table 1: **(Left)** MetaQA and **(Right)** WikiData Hits @1 for 1-3 hop sub-tasks. ots: off-the-shelf without re-training. †: obtained from Sun et al. (2019). cascade: adapted to multi-hop setting by repeatedly applying Eq. 2. pre: pre-trained on slot-filling. e2e: end-to-end trained on single-hop and multi-hop queries.

a very specific distribution of questions, and is not focused on entity- and relation-centric questions. Here we instead train the mention encoder using distant supervision from a KB.

Specifically, assume we are given an open-domain KB consisting of facts $(e_1, R, e_2)$ specifying that the relation $R$ holds between the subject $e_1$ and the object $e_2$. Then for a corpus of entity-linked text passages $\{d_k\}$, we automatically identify tuples $(d, (e_1, R, e_2))$ such that $d$ mentions both $e_1$ and $e_2$. Using this data, we learn to answer slot-filling queries in a reading comprehension setup, where the query $q$ is constructed from the surface form of the subject entity $e_1$ and a natural language description of $R$ (e.g. "Jerry Garcia, birth place, ?"), and the answer $e_2$ needs to be extracted from the passage $d$. Using string representations in $q$ ensures our pre-training setup is similar to the downstream task. In pretraining, we use the same scoring function as in previous section, but over all *spans* $m$ in the passage:

$$s(m, e_1, q) \propto \exp\{f(s) \cdot g(q, e_1)\}. \tag{6}$$

Following Seo et al. (2016), we normalize start and end probabilities of the span separately.

For effective transfer to the full corpus setting, we must also provide negative instances during pre-training, i.e. query and passage pairs where the answer is *not* contained in the passage. We consider three types of hard negatives: (1) *shared-entity negatives*, which pair a query $(e_1, R, ?)$ with a passage which mentions $e_1$ but not the correct tail answer; (2) *shared-relation negative*, which pair a query $(e_1, R, ?)$ with a passage mentioning two other entities $e_1'$ and $e_2'$ in the same relation $R$; and (3) *random negatives*, which pair queries with random passages from the corpus.

For the multi-hop slot-filling experiments below, we used WikiData (Vrandečić & Krötzsch, 2014) as our KB, Wikipedia as the corpus, and SLING (Ringgaard et al., 2017) to identify entity mentions. We restrict $d$ be from the Wikipedia article of the subject entity to reduce noise. Overall we collected $950K$ pairs over $550K$ articles. For the experiments with MetaQA, we supplemented this data with the corpus and KB provided with MetaQA, and string matching for entity linking.

## 3 EXPERIMENTS

### 3.1 METAQA: MULTI-HOP QUESTION ANSWERING WITH TEXT

**Dataset.** We first evaluate DrKIT on the MetaQA benchmark for multi-hop question answering (Zhang et al., 2018). MetaQA consists of around $400K$ questions ranging from 1 to 3 hops constructed by sampling relation paths from a movies KB (Miller et al., 2016) and converting them to natural language using templates. The questions cover $8$ relations and their inverses, around $43K$ entities, and are paired with a corpus consisting of $18K$ Wikipedia passages about those entities. The questions are all designed to be answerable using either the KB or the corpus, which makes it possible to compare the performance of our "virtual KB" QA system to a plausible upper bound system that has access to a complete KB. We used the same version of the data as Sun et al. (2019). Details of the implementation are in Appendix A.

**Results.** Table 1 shows the accuracy of the top-most retrieved entity (Hits@1) for the sub-tasks ranging from 1-3 hops, and compares to the state-of-the-art systems for the text-only setting on these

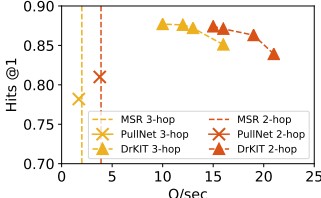 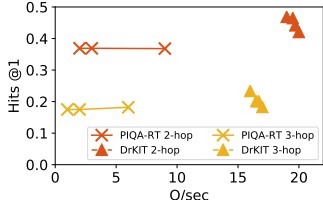 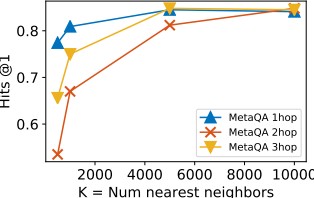

Figure 3: Hits @1 vs Queries/sec during inference on **(Left)** MetaQA and **(Middle)** WikiData tasks, measured on a single CPU server with 6 cores. MSR: Multi-step Retriever model from Das et al. (2019a) (we only show Q/sec). **(Right)** Effect of varying number of nearest neighbors $K$ during MIPS.

tasks. DrKIT outperforms the prior state-of-the-art by a large margin in the 2-hop and 3-hop cases. The strongest prior method, PullNet (Sun et al., 2019; 2018), uses a graph neural network model with learned iterative retrieval from the corpus to answer multi-hop questions. It uses the MetaQA KB during training to identify shortest paths between the question entity and answer entity, which are used to supervise the text retrieval and reading modules. DrKIT, on the other hand, has strong performance without such supervision, demonstrating its capability for end-to-end learning. (Adding the same intermediate supervision to DrKIT does not even consistently improve performance—it gives DrKIT a small lift on 1- and 2-hop questions but does not help for 3-hop questions.)

DrKIT's architecture is driven, in part, by efficiency considerations: unlike PullNet, it is designed to answer questions with minimal processing at query time. Figure 3 compares the tradeoffs between accuracy and inference time of DrKIT with PullNet as we vary $K$, the number of dense nearest neighbors retrieved. The runtime gains of DrKIT over PullNet range between 5x-15x.

**Analysis.** We perform ablations on DrKIT for the MetaQA data. First, we empirically confirm that taking a sum instead of max over the mentions of an entity hurts performance. So does removing the softmax temperature (by setting $\lambda = 1$). Removing the TFIDF component from Eq. 3, leads a large decrease in performance for 2-hop and 3-hop questions. This is because the TFIDF component *constrains* the end-to-end learning to be along reasonable paths of co-occurring mentions, preventing the search space from exploding. The results also highlight the importance of the pretraining method of §2.3, as DrKIT over an index of BERT rep-

| Ablations | 1hop | 2hop | 3hop |
|---|---|---|---|
| DrKIT | 0.844 | 0.860 | 0.876 |
| −Sum over $M_{z_t}$ | 0.837 | 0.823 | 0.797 |
| −$\lambda = 1$ | 0.836 | 0.752 | 0.799 |
| −w/o TFIDF | 0.845 | 0.548 | 0.488 |
| −BERT index | 0.634 | 0.610 | 0.555 |
| *Incomplete KB for pretraining* | | | |
| 25% KB | 0.839 | 0.804 | 0.830 |
| 50% KB | 0.843 | 0.834 | 0.834 |
| *(50% KB-only)* | *0.680* | *0.521* | *0.597* |

resentations without pretraining is 23 points worse in the 3-hop case. We also check the performance when the KB used for pre-training is *incomplete*. Even with only $50\%$ edges retained, we see good performance—better than PullNet and the state-of-the-art for a KB-only method (in *italics*).

We analyzed 100 2-hop questions correctly answered by DrKIT and found that for 83, the intermediate answers were also correct. The other 17 cases were all where the second hop asked about *genre*, e.g. "What are the genres of the films directed by Justin Simien?". We found that in these cases the intermediate answer was the same as the correct final answer—essentially the model learned to answer the question in 1 hop and copy it over for the second hop. Among incorrectly answered questions, the intermediate accuracy was only $47\%$, so the mistakes were evenly distributed across the two hops.

## 3.2 WikiData: Multi-Hop Slot-Filling

The MetaQA dataset has been fairly well-studied, but has limitations since it is constructed over a small KB. In this section we consider a new task, in a larger scale setting with many more relations, entities and text passages. The new dataset also lets us evaluate performance in a setting where the test set contains documents and entities not seen at training time, an important issue when devising a QA system that will be used in a real-world setting, where the corpus and entities in the discourse change over time, and lets us perform analyses not possible with MetaQA, such as extrapolating from single-hop to multi-hop settings without retraining.

**Dataset.**    We sample two subsets of Wikipedia articles, one for pre-training (§2.3) and end-to-end training, and one for testing. For each subset we consider the set of WikiData entities mentioned in the articles, and sample paths of 1-3 hop relations among them, ensuring that any intermediate entity has an in-degree of no more than $100$. Then we construct a semi-structured query by concatenating the surface forms of the head entity with the path of relations (e.g. "Helene Gayle, employer, founded by, ?"). The answer is the tail entity at the end of the path, and the task is to extract it from the Wikipedia articles. Existing slot-filling tasks (Levy et al., 2017; Surdeanu, 2013) focus on a *single-hop*, *static* corpus setting, whereas our task considers a *dynamic* setting which requires the system to traverse the corpus. For each setting, we create a dataset with $10K$ articles, $120K$ passages, $> 200K$ entities and $1.5M$ mentions, resulting in an index of size about 2gb. We include example queries in Appendix B.

**Baselines.**    We adapt two publicly available open-domain QA systems for this task – DrQA[3] (Chen et al., 2017) and PIQA[4] (Seo et al., 2019). While DrQA is relatively mature and widely used, PIQA is recent, and similar to our setup since it also answers questions with minimal computation at query time. It is broadly similar to a single textual follow operation in DrKIT, but is not constructed to allow retrieved answers to be converted to entities and then used in subsequent processing, so it is not directly applicable to multi-hop queries. We thus also consider a cascaded architecture which repeatedly applies Eq. 2, using either of PIQA or DrQA to compute $\Pr(z_t|q, z_{t-1})$ against the corpus, retaining at most $k$ intermediate answers in each step. We tune $k$ in the range of 1-10, since larger values make the runtime infeasible. Further, since these models were trained on natural language questions, we use the templates released by Levy et al. (2017) to convert intermediate questions into natural text.[5] We test off-the-shelf versions of these systems, as well as a version of PIQA re-trained on our our slot-filling data.[6] We compare to a version of DrKIT trained only on single-hop queries (§2.3) and similarly cascaded, and one version trained end-to-end on the multi-hop queries.

**Results.**    Table 1 (right) lists the Hits @1 performance on this task. Off-the-shelf open-domain QA systems perform poorly, showing the challenging nature of the task. Re-training PIQA on the slot-filling data improves performance considerably, but DrKIT trained on the same data improves on it. A large improvement over these cascaded architectures is seen with end-to-end training, which is made possible by the differentiable operation introduced in this paper. We also list the performance of DrKIT when trained against an index of fixed BERT-large mention representations. While this is comparable to the re-trained version of PIQA, it lags behind DrKIT pre-trained using the KB, once again highlighting the importance of the scheme outlined in §2.3. We also plot the Hits @1 against Queries/sec for cascaded versions of PIQA and DrKIT in Figure 3 (middle). We observe runtime gains of 2x-3x to DrKIT due to the efficient implementation of entity-mention expansion of §2.2.

**Analysis.**    In order to understand where the accuracy gains for DrKIT come from, we conduct experiments on the dataset of slot-filling queries released by Levy et al. (2017). We construct an *open* version of the task by collecting Wikipedia articles of all subject entities in the data. A detailed discussion is in Appendix C, and here we note the main findings. PIQA trained on SQuAD only gets $30\%$ macro-avg accuracy on this data, but this improves to $46\%$ when re-trained on our slot-filling data. Interestingly, a version of DrKIT which selects from *all spans* in the corpus performs similarly to PIQA ($50\%$), but when using entity linking it significantly improves to $66\%$. It also has $55\%$ accuracy in answering queries about *rare* relations, i.e. those observed $< 5$ times in its training data. We also conduct probing experiments comparing the representations learned using slot-filling to those by vanilla BERT. We found that while the two are comparable in detecting *fine-grained entity types*, the slot-filling version is significantly better at encoding *entity co-occurrence* information.

### 3.3    HOTPOTQA: MULTI-HOP INFORMATION RETRIEVAL

**Dataset.**    HotpotQA (Yang et al., 2018) is a recent dataset of over 100K crowd-sourced multi-hop questions and answers over introductory Wikipedia passages. We focus on the open-domain *fullwiki*

---

[3] https://github.com/facebookresearch/DrQA
[4] https://github.com/uwnlp/denspi
[5] For example, "Helene Gayle. employer?" becomes "Who is the employer of Helene Gayle?"
[6] We tuned several hyperparameters of PIQA on our data, eventually picking the *sparse first* strategy, a sparse weight of $0.1$, and a filter threshold of $0.2$. For the SQuAD trained version, we also had to remove paragraphs smaller than 50 tokens since with these the model failed completely.

| Model | Q/s | Accuracy | | | |
|---|---|---|---|---|---|
| | | @2 | @5 | @10 | @20 |
| BM25[†] | – | 0.093 | 0.191 | 0.259 | 0.324 |
| PRF-Task[†] | – | 0.097 | 0.198 | 0.267 | 0.330 |
| BERT re-ranker[†] | – | 0.146 | 0.271 | 0.347 | 0.409 |
| Entity Centric IR[†] | 0.32[*] | 0.230 | 0.482 | 0.612 | 0.674 |
| DrKIT (WikiData) | | 0.355 | 0.588 | 0.671 | **0.710** |
| DrKIT (Hotpot) | **4.26**[*] | **0.385** | 0.595 | 0.663 | 0.703 |
| DrKIT (Combined) | | 0.383 | **0.603** | **0.672** | **0.710** |

| Model | EM | F1 |
|---|---|---|
| Baseline[†] | 0.288 | 0.381 |
| +EC IR[‡] | 0.354 | 0.462 |
| +Golden Ret[◇] | **0.379** | **0.486** |
| +DrKIT[†] | 0.357 | 0.466 |

Table 2: **(Left)** Retrieval performance on the HotpotQA benchmark dev set. Q/s denotes the number of queries per second during inference on a single 16-core CPU. Accuracy @$k$ is the fraction where *both* the correct passages are retrieved in the top $k$. [†]: Baselines obtained from Das et al. (2019b). For DrKIT, we report the performance when the index is pretrained using the WikiData KB alone, the HotpotQA training questions alone, or using both. [*]: Measured on different machines with similar specs. **(Right)** Overall performance on the HotpotQA task, when passing 10 retrieved passages to a downstream reading comprehension model (Yang et al., 2018). [‡]: From Das et al. (2019b). [◇]: From Qi et al. (2019). [†]: Results on the dev set.

setting where the two gold passages required to answer the question are not known in advance. The answers are free-form spans of text in the passages, not necessarily entities, and hence our model which selects entities is not directly applicable here. Instead, inspired by recent works (Das et al., 2019b; Qi et al., 2019), we look at the challenging sub-task of *retrieving* the passages required to answer the questions from a pool of 5.23M. This is a multi-hop IR task, since for many questions at least one passage may be 1-2 hops away from the entities in the question. Further, each passage is about an entity (the title entity of that Wikipedia page), and hence retrieving passages is the same as identifying the title entities of those passages. We apply DrKIT to this task of identifying the two entities for each question, whose passages contain the information needed to answer that question. Then we pass the top 10 passages identified this way to a standard reading comprehension architecture from Yang et al. (2018) to select the answer span.

**Setup.** We use the Wikipedia abstracts released by Yang et al. (2018) as the text corpus.[7] The total number of entities is the same as the number of abstracts, 5.23M, and we consider hyperlinks in the text as mentions of the entities to whose pages they point to, leading to 22.8M total mentions in an index of size 34GB. For pretraining the mention representations, we compare using the WikiData KB as described in §2.3 to directly using the HotpotQA training questions, with TFIDF based retrieved passages as negative examples. We set $A_{E \to M}[e, m] = 1$ if either the entity $e$ is mentioned on the page of the entity denoted by $m$, or vice versa. For entity linking over the questions, we retrieve the top 20 entities based on the match between a bigram based TFIDF vector of the question with a similar vector derived from the surface form of the entity (same as the title of the Wiki article). We found that the gold entities that need to be retrieved are within 2 hops of the entities linked in this manner for 87% of the dev examples.

Unlike the MetaQA and WikiData datasets, however, for HotpotQA we do not know the number of hops required for each question in advance. Instead, we run DrKIT for 2 hops for each question, and then take a weighted average of the distribution over entities after each hop $Z^* = \pi_0 Z_0 + \pi_1 Z_1 + \pi_2 Z_2$. $Z_0$ consists of the entities linked to the question itself, rescored based on an encoding of the question, since in some cases one or both the entities to be retrieved are in this set.[8] $Z_1$ and $Z_2$ are given by Eq. 4. The mixing weights $\pi_i$ are the softmax outputs of a classifier on top of another encoding of the question, learnt *end-to-end* on the retrieval task. This process can be viewed as soft mixing of different templates ranging from 0 to 2 hops for answering a question, similar to NQL (Cohen et al., 2019).

**Results.** We compare our retrieval results to those presented in Das et al. (2019b) in Table 2 (Left). We measure the accuracy @$k$ retrievals, which is the fraction of questions for which *both* the required passages are in the top $k$ retrieved ones. We see an improvement in accuracy across the board,

---

[7] https://hotpotqa.github.io/wiki-readme.html

[8] For example, for the question "How are elephants connected to Gajabrishta?", one of the passages to be retrieved is "Gajabrishta" itself.

| System | Runtime | | Answer | | Sup Fact | | Joint | |
|---|---|---|---|---|---|---|---|---|
| | #Bert | s/Q | EM | F1 | EM | F1 | EM | F1 |
| Baseline (Yang et al., 2018) | – | – | 25.23 | 34.40 | 5.07 | 40.69 | 2.63 | 17.85 |
| Golden Ret (Qi et al., 2019) | – | 1.4[†] | 37.92 | 48.58 | 30.69 | 64.24 | 18.04 | 39.13 |
| Semantic Ret (Nie et al., 2019) | 50* | 40.0[‡] | 45.32 | 57.34 | 38.67 | 70.83 | 25.14 | 47.60 |
| HGN (Fang et al., 2019) | 50* | 40.0[‡] | 56.71 | 69.16 | **49.97** | 76.39 | **35.63** | 59.86 |
| Rec Ret (Asai et al., 2020) | 500* | 133.2[†] | **60.04** | **72.96** | 49.08 | **76.41** | 35.35 | **61.18** |
| DrKIT + BERT | **1.2[◇]** | **1.3** | 42.13 | 51.72 | 37.05 | 59.84 | 24.69 | 42.88 |

Table 3: Official leaderboard evaluation on the test set of HotpotQA. #Bert refers to the number of calls to BERT (Devlin et al., 2019) in the model. s/Q denotes seconds per query (using batch size 1) for inference on a single 16-core CPU. Answer, Sup Fact and Joint are the official evaluation metrics for HotpotQA. *: This is the minimum number of BERT calls based on model and hyperparameter descriptions in the respective papers. [†]: Computed using code released by authors, using a batch size of 1. [‡]: Estimated based on the number of BERT calls, using 0.8s as the time for one call (without considering overhead due to other computation in the model). [◇]: One call to a 5-layer Transformer, and one call to BERT.

with much higher gains @2 and @5. The main baseline is the entity-centric IR approach which runs a BERT-based re-ranker on 200 pairs of passages for each question. Importantly, DrKIT also improves by over 10x in terms of queries per second during inference. Note that the inference time is measured using a batch size of 1 for both models for fair comparison. DrKIT can be easily run with batch sizes up to 40, but the entity centric IR baseline cannot due to the large number of runs of BERT for each query. When comparing different datasets for pretraining the index, there is not much difference between using the WikiData KB, or the HotpotQA questions. The latter has a better accuracy @2, but overall the best performance is when using a combination of both.

In Table 2 (Right), we check the performance of the baseline reading comprehension model from Yang et al. (2018), when given the passages retrieved by DrKIT. While there is a significant improvement over the baseline which uses a TFIDF based retrieval, we see only a small improvement over the passages retrieved by the entity-centric IR baseline, despite the significantly improved accuracy @10 of DrKIT. Among the 33% questions where the top 10 passages do not contain both the correct passages, for around 20% the passage containing the answer is also missing. We conjecture this percentage is lower for the entity-centric IR baseline, and the downstream model is able to answer some of these questions without the other supporting passage.

Lastly, we feed the top 5 passages retrieved by DrKIT to an improved answer span extraction model based on BERT. This model implements a standard architecture for extracting answers from text, and is trained to predict both the answers and the supporting facts. Details are included in Appendix D. Table 3 shows the performance of this system on the HotpotQA test set, compared with other recently published models on the leaderboard.[9] In terms of accuracy, DrKIT+BERT reaches a modest score of 42.88 joint F1, but is considerably faster (up to 100x) than the models which outperform it.

## 4 RELATED WORK

Neural Query Language (NQL) (Cohen et al., 2019) defines differentiable templates for multi-step access to a symbolic KB, in which relations between entities are *explicitly* enumerated. Here, we focus on the case where the relations are implicit in mention representations derived from text. Knowledge Graph embeddings (Bordes et al., 2013; Yang et al., 2014; Dettmers et al., 2018) attach continuous representations to discrete symbols which allow them to be incorporated in deep networks (Yang & Mitchell, 2017). Embeddings often allow generalization to unseen facts using relation patterns, but text corpora are more complete in the information they contain.

Talmor & Berant (2018) also examined answering compositional questions by treating a text corpus (in their case the entire web) as a KB. However their approach consists of parsing the query into a computation tree, and running a black-box QA model on its leaves separately, which *cannot* be trained end-to-end. Recent papers have also looked at complex QA using graph neural networks

---

[9] As of February 23, 2020: `https://hotpotqa.github.io/`.

(Sun et al., 2018; Cao et al., 2019; Xiao et al., 2019) or by identifying paths of entities in text (Jiang et al., 2019; Kundu et al., 2019; Dhingra et al., 2018). These approaches rely on identifying a small relevant pool of evidence documents containing the information required for multi-step QA. Hence, Sun et al. (2019) and Ding et al. (2019), incorporate a dynamic retrieval process to add text about entities identified as relevant in the previous layer of the model. Since the evidence text is processed in a query-dependent manner, the inference speed is slower than when it is pre-processed into an indexed representation (see Figure 3). The same limitation is shared by methods which perform multi-step retrieval interleaved with a reading comprehension model (Das et al., 2019a; Feldman & El-Yaniv, 2019; Lee et al., 2019).

## 5 CONCLUSION

We present DrKIT, a differentiable module that is capable of answering multi-hop questions directly using a large entity-linked text corpus. DrKIT is designed to imitate traversal in KB over the text corpus, providing ability to follow relations in the "virtual" KB over text. We achieve state-of-the-art results on the MetaQA dataset for answering natural language questions, with a 9 point increase in the 3-hop case. We also developed an efficient implementation using sparse operations and inner product search, which led to a 10-100x increase in Queries/sec over baseline approaches.

ACKNOWLEDGMENTS

Bhuwan Dhingra was supported by a Siemens fellowship during this project. This work was supported in part by ONR Grant N000141812861, Google, Apple, and grants from NVIDIA.

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

## A  METAQA: IMPLEMENTATION DETAILS

We use $p = 400$ dimensional embeddings for the mentions and queries, and 200-dimensional embeddings each for the start and end positions. This results in an index of size 750MB. When computing $A_{E \to M}$, the entity to mention co-occurrence matrix, we only retain mentions in the top 50 paragraphs matched with an entity, to ensure sparsity. Further we initialize the first 4 layers of the question encoder with the Transformer network from pre-training. For the first hop, we assign $Z_0$ as a 1-hot vector for the least frequent entity detected in the question using an exact match. The number of nearest neighbors $K$ and the softmax temperature $\lambda$ were tuned on the dev set of each task, and we found $K = 10000$ and $\lambda = 4$ to work best. We pretrain the index on a combination of the MetaQA corpus, using the KB provided with MetaQA for distance data, and the WikiData corpus.

## B  WIKIDATA DATASET STATISTICS

| Task | #train | #dev | #test | $|\mathcal{E}_{test}|$ | $|\mathcal{M}_{test}|$ | $|\mathcal{D}_{test}|$ | Example |
|---|---|---|---|---|---|---|---|
| 1hop | 16901 | 2467 | 10000 | 216K | 1.2M | 120K | Q. Mendix, industry?
A. Enterprise Software |
| 2hop | 163607 | 398 | 9897 | 342K | 1.9M | 120K | Q. 2000 Hel van het Mergelland, winner, place of birth?
A. Bert Grabsch → Lutherstadt Wittenberg |
| 3hop | 36061 | 453 | 9899 | 261K | 1.8M | 120K | Q. Magnificent!, record label, founded by, date of death?
A. Prestige → Bob Weinstock → 14 Jan 2006 |

Table 4: WikiData dataset

Details of the collected WikiData dataset are shown in Table 4.

## C  INDEX ANALYSIS

**Single-hop questions and relation extraction.**   Levy et al. (2017) released a dataset of $1M$ slot-filling queries of the form $(e_1, R, ?)$ paired with Wikipedia sentences mentioning $e_1$, which was used for training systems that answered single-step slot-filling questions based on a small set of candidate passages. Here we consider an *open* version of the same task, where answers to the queries must be extracted from a corpus rather than provided candidates. We construct the corpus by collecting and entity-linking all paragraphs in the Wikipedia articles of all $8K$ subject entities in the dev and test sets, leading to a total of $109K$ passages. After constructing the TFIDF $A_{E \to M}$ and coreference $B_{M \to E}$ matrices for this corpus, we directly use our pre-trained index to answer the test set queries.

Figure 4 (Right) shows the Hits@1 performance of the Levy et al. (2017) slot-filling dataset. We report results on 2 subsets of relations in addition to all relations. The Rare subset comprises of relations with frequencies $< 5$ in the training data while the 'Frequent' subset contains the rest. DrKIT on entity-mentions consistently outperforms the other phrase-based models showing the benefit of

| Probing Task | Negative Example | BERT | DrKIT |
|---|---|---|---|
| Shared Entity | Neil Herron played for West of Scotland. | 0.850 | 0.876 |
| Shared Relation | William Paston was a British politician. | 0.715 | 0.846 |

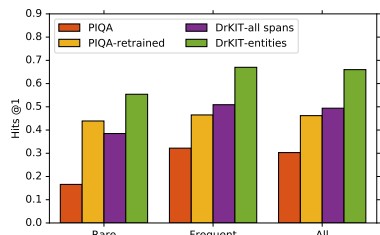

Figure 4: **Left:** F1 scores on Shared Entity and Shared Relation negatives. The negative examples are for the Query : (Neil Herron, occupation, ?). **Right:** Macro-avg accuracy on the Levy et al. (2017) relation extraction dataset. We split the results based on frequency of the relations in our WikiData training data. DrKIT-all spans refers to a variant of our model which selects from all spans in the corpus, instead of only entity-linked mentions.

indexing only entity-mentions in single-hop questions over all spans. Note that DrKit-entities has a high Hits@1 performance on the Rare relations subset, showing that there is generalization to less frequent data due to the natural language representations of entities and relations.

**Probing Experiments** Finally, to compare the representations learned by the BERT model fine-tuned on the WikiData slot-filling task, we design two probing experiments. In each experiment, we keep the parameters of the BERT model (mention encoders) being probed fixed and only train the query encoders. Similar to Tenney et al. (2019), we use a weighted average of the layers of BERT here rather than only the top-most layer, where the weights are learned on the probing task.

In the first experiment, we train and test on shared-entity negatives. Good performance here means the BERT model being probed encodes fine-grained *entity-type* information reliably[10]. As shown in Table 4, BERT performs well on this task, suggesting it encodes fine-grained types well.

In the second experiment, we train and test only on shared-relation negatives. Good performance here means that the BERT model encodes *entity co-occurrence* information reliably. In this probe task, we see a large performance drop for BERT, suggesting it does not encode entity co-occurrence information well. The good performance of the DrKIT model on both experiments suggests that fine-tuning on the slot-filling task primarily helps the contextual representations to also encode entity co-occurrence information, in addition to entity type information.

## D HOTPOTQA ANSWER EXTRACTION

On HotpotQA, we use DrKIT to identify the top passages which are likely to contain the answer to a question. We then train a separate model to extract the answer from a concatenation of these passages. This model is a standard BERT-based architecture used for SQuAD (see Devlin et al. (2019) for details), with a few modifications. First, to handle boolean questions, we train a 3-way classifier on top of the [CLS] representation from BERT to decide whether the question has a "span", "yes" or "no" answer, respectively. During inference, if this classifier has the highest probability on "span" we extract a start and end position similar to Devlin et al. (2019), else we directly answer as "yes" or "no".

Second, to handle supporting fact prediction, we prepend each sentence in the concatenated passages passed to BERT with a special symbol [unused0], and train a binary classifier on top of the representation of each of these symbols output from BERT. The binary classifier is trained to predict 1 for sentences which are supporting facts and 0 for sentences which are not. During inference, we take all sentences for which the output probability of this classifier is $> 0.5$ as supporting facts.

The training loss is an average of the loss for the 3-way classifier ($\mathcal{L}_{cls}$), the sum of the losses for the supporting fact classifiers ($\mathcal{L}_{sp}$), and the losses for the start and end positions of span answers ($\mathcal{L}_{st}, \mathcal{L}_{en}$):

$$\mathcal{L} = (\mathcal{L}_{cls} + \mathcal{L}_{sp} + \mathcal{L}_{st} + \mathcal{L}_{en})/4 \tag{7}$$

---

[10]A reasonable heuristic for solving this task is to simply detect an entity with the correct type in the given sentence, since all sentences contain the subject entity.

We train the system on $5$ passages per question, provided in the distractor setting of HotpotQA—
2 gold ones and 3 negatives from a TFIDF retriever. We keep the gold passages at the beginning
for $60\%$ of the examples, and randomly shuffle all passages for the rest, since during inference the
correct passages are likely to be retrieved at the top by DrKIT. Other hyperparameters include—
batch size $32$, learning rate $5 \times 10^{-5}$, number of training epochs $5$, and a maximum combined
passage length $512$.

