# OpenReview forum: "Differentiable Reasoning over a Virtual Knowledge Base"
_ICLR.cc/2020/Conference — Accept (Talk)_

### Official Review · AnonReviewer2 · 2019-10-20
**Official Blind Review #2**

**Rating:** 8

**Review:**

This paper introduces a new architecture for question answering, that can be trained in an end-to-end fashion. The model expands the entities in a given question to relevant mentions, which are in turn aggregated to a new set of entities. The procedure can be repeated for multi-hop questions. The resulting entities are returned as candidate answers. The approach relies on an index of mentions for approximate MIPS, and on sparse matrix-vector products for fast computation. Overall, the model processes queries 10x faster than previous approaches. The method provides state-of-the-art results on the MetaQA benchmark, with significant improvements on 3-hop questions. The experiments are detailed, and the paper is very well written.

A few comments / questions:

1. Do you have any explanation of why taking the max instead of the sum has a significant impact on the 2,3-hop performance, but only gives a small improvement for 1-hop questions?

2. The same observation can be done for the temperature lambda=1 vs lambda=4, so I was wondering about the distribution of the entities you get on the output of the softmax (in Eq 4). Is the distribution very spiky, and Z_t usually only composed of a few entities? In that case, I guess lambda=4 encourages the model to explore/update more relation paths? Something that works very well in text generation is to not just set a temperature, but also to apply a softmax only on the K elements with the highest score (so the softmax is applied on K logits, and everything else is set to 0). Did you consider something like this? It may prevent the model from considering irrelevant entities, but also from considering only a few ones.

3. Given the iterative procedure of the method, I wonder how well the model would generalize to more hops. Did you try for instance to train with 1/2 hops and test whether it can generalize to 3-hop questions?

4. In Section 2.3, you fix the mention encoder because training the encoder would not work with the approximate nearest neighbor search (I assume this is because the index would need to be rebuilt). However, the ablation study suggests that the pretraining is critical, and one could imagine that fine-tuning the mention encoder would improve the performance even further. Instead of considering a mention encoder, could you have a lookup table of mentions (initialized with BERT applied on each mention), where mention embeddings are fine-tuned during training? The problem of the index is still there, but could you consider an exact search on the lookup table (exact search over a few million embeddings is slow, but it should still run in a reasonable amount of time using a framework like FAISS, and it would give an upper-bound of the performance you could achieve by fine-tuning the mention encoder).

**Experience Assessment:**

I have read many papers in this area.

**Review Assessment: Checking Correctness Of Derivations And Theory:**

I assessed the sensibility of the derivations and theory.

**Review Assessment: Checking Correctness Of Experiments:**

I assessed the sensibility of the experiments.

**Review Assessment: Thoroughness In Paper Reading:**

I read the paper at least twice and used my best judgement in assessing the paper.

---

> ### Author Response · Authors · 2019-11-13
> **Response to AnonReviewer2**
>
> We thank the reviewer for the valuable feedback. We have updated the paper draft to reflect the feedback. In particular we have added section 3.3 with new results on HotpotQA.
>
> 1. Regarding max instead of sum:
>
> For multi-hop questions, the aggregated score of an entity is used as input to a softmax function to compute Z_t, which is the input distribution over entities for the next hop. If we take a sum over all mentions, entities which appear multiple times in the retrieved set will tend to have a much higher score than those which don’t, and end up with a score close to 1 in Z_t. This prevents the model from exploring multiple relation paths, same as the case you point out in 2.
>
> 2. Regarding the softmax temperature:
>
> You are correct for the reason why lambda=4 helps for multi-hop questions. We did not consider the alternative strategy of applying softmax only on the top K elements, but we note that often the score difference between the top and the second best mentions is already high (recall that these are maximum inner product scores against a fixed index). In this case, pruning the set of entities over which the softmax is applied would not help.
>
> 3. Regarding generalization to more hops:
>
> The `DrKIT (pre, cascade)` model in Table 1 (right) is exactly this model. It is only trained on 1-hop questions, but cascaded for 2-hop and 3-hop questions as well. We find that it is significantly better than similarly cascaded versions of PIQA and DrQA, but worse than a version of DrKIT which is fine-tuned end-to-end.
>
> 4. Regarding updating the mention embeddings:
>
> This is a very interesting suggestion. However, note that our setup assumes that the mentions involved in test questions are different from those involved at training time. For WikiData, the test corpus itself is a different subset of Wikipedia, whereas for MetaQA many answers to test questions are not part of training questions. In this case, updating a subset of mention embeddings which are only relevant for the training questions could lead the model to perform worse on the mentions relevant at test time, whose embeddings are left to the pre-trained values.
>
> In a different scenario, where the training and test questions are over the same set of mentions, we agree that updating mention embeddings might lead to further improvements.

---

> > ### Comment · AnonReviewer2 · 2019-11-15
> > **Thank you for the feedback**
> >
> > I thank the authors for their responses. I'm satisfied with the rebuttal, and will stand by my positive rating.

---

### Official Review · AnonReviewer1 · 2019-10-21
**Official Blind Review #1**

**Rating:** 8

**Review:**

The paper proposes a model that can perform multi-hop question-answering based on a textual knowledge base. Results show that the proposed model -- DrKIT -- performed close to or better than the state of the art results over MetaQA and WikiData datasets. Ablation study is offered to show that a few tricks in the model are necessary to make it work, and comparisons with baseline models such as DrQA and PIQA are presented. The paper also provides additional means to speed up the DrKIT model using the hashing trick and approximated top-k methods.

The paper is a good one and I vote for its acceptance. Besides achieving good performance, the proposed DrKIT model makes sense, and all the parts are necessary components based on the ablation study results. In addition, the ablation study and the speed-up methods are great addition to the model to make it work better.

With this generally positive assessment said, I do have a few questions below that I hope the authors could provide some response. These are on top of the high quality of the paper, and should be best regarded as suggestions for future work.

1. In equation (3), do G and F have to be TFIDF features? The likes of word2vec and GloVe (and also pershaps fastText) are trained based on co-occurences of adjacent words, and I would imagine that they will improve over TFIDF. This is just an intuition and I could be wrong, but it would be very helpful to hear the authors' opinions.

2. The ablation study mentioned that the softmax temperature helps with the model. This is a nice observation, but is there any intuition behind why that is the case? I could imagine that it could be because the gradients of a saturated softmax function is small and therefore results in slow training of the model. If this is the case, both low temperature and high temperature will fail to work. It would have been better so show both ends of failing extremes in an ablation study.

3. Can you discuss the similarity between DrKIT and multi-hop End-to-End Memory Networks [1]? It looks very much like an expansion of it with a fixed retrieval mechanism and by expanding the answer to a set rather than a single vector.

[1] Sainbayar Sukhbaatar, Arthur Szlam, Jason Weston, Rob Fergus, End-To-End Memory Networks, NIPS 2015


**Experience Assessment:**

I have published one or two papers in this area.

**Review Assessment: Checking Correctness Of Derivations And Theory:**

I assessed the sensibility of the derivations and theory.

**Review Assessment: Checking Correctness Of Experiments:**

I assessed the sensibility of the experiments.

**Review Assessment: Thoroughness In Paper Reading:**

I read the paper at least twice and used my best judgement in assessing the paper.

---

> ### Author Response · Authors · 2019-11-13
> **Response to AnonReviewer1**
>
> We thank the reviewer for the valuable feedback. We have updated the paper draft to reflect the feedback. In particular we have added section 3.3 with new results on HotpotQA.
>
> 1. Regarding G and F in Eq. (3):
>
> The purpose for the first term in Eq. (3) is to have a sparse mapping from an entity to the mentions related to that entity. Indeed, the reviewer correctly points out that there is no need for this mapping to be computed using TF-IDF features. Any of the latest techniques from the IR literature can be used to compute this fixed mapping, and this is definitely an area for future work to investigate. We use TF-IDF since these are easy to compute, highly scalable, and work pretty well in practice.
>
> 2. Regarding the softmax temperature:
>
> Since we are using a maximum inner product search to retrieve the top-K mentions, the relevance scores s(m, z_{t-1}, q) for these are all usually high. Using a softmax function directly on top of these scores leads to a very peaked distribution, and for multi-hop questions, this results in effectively only one entity being passed from one hop to the next. The softmax temperature flattens out this distribution, so the model can explore many paths from one hop to the next.
>
> 3. Regarding DrKIT and end-to-end Memory Networks:
>
> The key difference between our work and memory networks, is that in our case the memories (or the index) is grounded in actual textual mentions from a potentially large text corpus. This results in a much larger size of the memory as compared to what was used in [1]. This necessitates the use of pretraining to learn their representations, and a MIPS operation to retrieve from them, which was not explored in [1]. Further, our retrieval from the index is a set of entities rather than a high-dimensional vector, which allows us to combine the strengths of sparse and dense retrieval strategies.

---

> > ### Comment · AnonReviewer1 · 2019-11-13
> > **Great feedback**
> >
> > Thanks to the authors to provide a feedback to the review. It answered the questions well, and I noticed the relevant paragraphs in the paper that incorporated them. I would like to maintain the positive assessment.

---

### Official Review · AnonReviewer3 · 2019-10-23
**Official Blind Review #3**

**Rating:** 8

**Review:**

The paper studies scaling multi-hop QA to large document collections, rather than working with small candidate lists of  document/paragraphs  (as done in most of the previous work), a very important, practical and challenging direction.

They start with linking mentions to entities in a knowledge base. Every iteration of their mult-hop system produces a set of entities Z_t, relying on entities predicted on the first representation Z_{t-1} and the question representation.   In order to make training tractable, they mask 'attention'  between Z_{t-1}  and Z_t (actually mentions corresponding to Z_t).  They also use top-K relevant mentions at train and test time. As the attention score is based on dot-product, they can plug-in the approximate Maximum Inner Product Search to avoid computing the attention score for every mention in the collection. The architecture is essentially end-to-end trainable (except for specialized pretraining discussed below).

Whereas itself the architecture is not overly novel (e.g., the architecture does feel a lot similar to models in KB context, and also graph convolution networks applied to QA), there is a lot of clever engineering and the novelty is really in showing that it can work without candidate preselection.

My main worry is pretraining. In both experiments (MetaQA and the new Wikidata Slot Filling task), they pretrain the encoders using a knowledge base, and the knowledge base directly corresponds to the QA task.   E.g., for MetaQA the questions are answerable using the knowledge bases, so relation types in the knowledge base presumably correspond to relations that need to be captured in the QA multihop learning.  This specialized pretraining appears to be crucial (88% for pretraining vs 55% with BER), presumably because of the top K pruning. Though a nice trick, it is likely limiting as a knowledge base needs to be available + it probably constraints the types of mult-hop questions the apporach can handle.  Also, some of the baselines do not benefit from using the KB, and, in principle, if it is used in training, why not use the KB at test time?  (I see though that for the second dataset pretraining seems to be done on a different part of Wikipedia, I guess, to address these concerns).

I was not sure how the number of hops T was selected for the model, it does not seem to be defined in the paper. Do you pretend that you know the true number of hops for each given question?

The authors experiment with reducing the size of a KB for pretraining. It apparently does not harm the first 1 hop questions, but 2 and 3-hop. Do the authors have any explanation for this?  Related to the previous question, does it mean that the model does not learn to exploit the hops for t > 1?

The evaluation is on MetaQA and on the newly introduced Wikidata task, whereas most (?) recent multi-hop QA work has focused on HotpotQA (and to certain degree WikiHop). Is the reason for not using (additionally) HotpotQA?  Is the model suitable for HotpotQA? If not, does this have to do with pretraining or the types of questions in HotpotQA?

The model definition in section 2.1 is not very easy to follow. E.g., it is not immediately clear if the model applied at every hop is the same model, and not clear how the model is made aware of the current search state (e.g., which part of the question has already processed / how the history is encoded) or even of the hop id.

I would really like to see more analysis of what the model learns at every hop.

--
After the rebuttal -- I appreciate the detailed feedback, extra experiments, and analysis. `I increased my score.


**Experience Assessment:**

I have published one or two papers in this area.

**Review Assessment: Checking Correctness Of Derivations And Theory:**

I assessed the sensibility of the derivations and theory.

**Review Assessment: Checking Correctness Of Experiments:**

I carefully checked the experiments.

**Review Assessment: Thoroughness In Paper Reading:**

I read the paper at least twice and used my best judgement in assessing the paper.

---

> ### Author Response · Authors · 2019-11-13
> **Response to AnonReviewer3**
>
> We thank the reviewer for the valuable feedback.
>
> ====> Regarding evaluation on HotpotQA <====
>
> We have added additional results in a new Section 3.3. The model is not directly applicable to HotpotQA since answers in that dataset are not necessarily entities. However, following very recent work [Godbole, 2019] we have identified a sub-task which is suitable for our model — retrieving the two introductory passages from Wikipedia which are required to answer a question. Each passage is associated with the title entity of that page, so the task boils down to selecting two entities given the question. Further, this task is multi-hop, since in many cases the entities to be retrieved are related to a question entity via a path of implicit relations.
>
> In section 3.3, we show that our approach outperforms [Godbole, 2019] by 6 points accuracy @10, the main metric for measuring retrieval performance. We also see more than 10x improvement in terms of inference time. When the retrieved passages are fed to a baseline reading comprehension model, we see an improvement of 8.5 F1 over a TF-IDF based retrieval, and we conjecture this can be further improved by using a more sophisticated reading comprehension model.
>
> [Godbole, 2019] Multi-step Entity-centric Information Retrieval for Multi-Hop Question Answering. EMNLP, 2019.
>
> ====> Regarding the importance of pretraining <====
>
> The reviewer raises an important point about the limitations posed by pre-training on the KB.
>
> Regarding the point about using the KB at test time, we think the model could conceptually be extended to use KB triples as well as text, but in this paper we focused on QA from text alone. It is important to verify, however, that the model is reading the text in some generalizable way, not just memorizing the KB triples. We note that in our ablation study we compare using 50% KB for pre-training vs using it directly at test time for answering questions. The former is between 20-30% better across the 3 hops, suggesting that the pre-training allows generalization beyond the facts present in the KB. Furthermore, as the reviewer points out, on Wikidata we test on an entirely different subset of entities than used for pretraining, so in this case using the KB at test would provide no improvements.
>
> Additionally, we have added additional results for multi-hop information retrieval on the HotpotQA dataset in section 3.3. Part of these experiments compare using the WikiData KB vs. the HotpotQA questions themselves for pretraining. While the HotpotQA based pretraining is better in accuracy @2, they are both very similar in accuracy @10, and both are significantly better than the best baseline of [Godbole, 2019]. Since, the construction of this dataset was not based in any way on the WikiData KB, the fact that the KB based pre-training works well for it suggests that it may be generally applicable for many types of multi-hop questions. A more detailed investigation of this aspect is, however, beyond the scope of this paper.
>
> ====> Regarding how the number of hops T was selected for the model <====
>
> Yes, following previous work, we assume the number of hops to be known for MetaQA and WikiData experiments.
>
> In cases where this cannot be determined in advance, we can use a soft mixture of the outputs after each hop. This is the case for the newly added HotpotQA results, where we used an additional classifier (trained end-to-end with the other parameters of the model) to determine the number of hops needed for each question softly). Please see section 3.3 for details.
>
> ====> Regarding reducing the size of KB for pretraining <====
>
> We believe reducing the size of the KB hurts because the quality of the pre-trained index reduces. This hurts 2-hop and 3-hop questions more because of error cascading -- with more hops there are more retrieval steps against the index.
>
> ====> Regarding the model definition in 2.1 <====
>
> The model architecture is identical at each hop, but the query representations used for retrieval are different. This results in a different relevance scoring function for each hop, and we have updated the paper to reflect this better (using s_t instead of s to denote the scoring function). Note that implicitly, the scoring function picks out the mentions which satisfy the relation requested for that hop, with an entity output by the previous hop (through z_{t-1}).
>
> ====> Regarding analysis of each hop <====
>
> We have added analysis of the intermediate predictions made by the model for 2-hop questions in MetaQA in section 3.1 (Analysis). For 100 correctly answered questions, we found that for 83 the intermediate answers were also correct. In the other 17 cases, the intermediate answer was the same as the final answer — essentially the model learned to answer the question in 1 hop and copy it over for the second hop. Among incorrectly answered questions, the intermediate accuracy is only 47%, so the mistakes are evenly distributed across the two hops.

---

> > ### Comment · AnonReviewer3 · 2019-11-13
> > **thanks for the feedback**
> >
> > Thank you for the extra analysis, new experiments and clarifications!  I decided to increase the (already positive) score.

---

### Decision · Program_Chairs · 2019-12-19

**Decision:**

Accept (Talk)

**Comment:**

This paper proposes a novel architecture for question-answering, which is trained in an end-to-end fashion.

The reviewers were unanimous in their vote to accept. Authors are encouraged to revise addressing reviewer comments.